# LPFQA: A Long-Tail Professional Forum-based Benchmark for LLMs' Evaluation

## Abstract

Large Language Models (LLMs) have made rapid progress in reasoning, question answering, and professional applications; however, their true capabilities remain difficult to evaluate using existing benchmarks. Current datasets often focus on simplified tasks or artificial scenarios, overlooking long-tail knowledge and the complexities of real-world applications. To address this gap, we propose LPFQA, a benchmark derived from authentic professional forums across 20 academic and industrial fields, covering 502 tasks grounded in practical expertise. LPFQA introduces four key innovations: fine-grained evaluation dimensions that target knowledge depth, reasoning, terminology comprehension, and contextual analysis; a hierarchical difficulty structure that ensures semantic clarity and unique answers; authentic professional scenario modeling with realistic user personas; and interdisciplinary knowledge integration across diverse domains. We evaluated 12 mainstream LLMs on LPFQA and observed significant performance disparities, especially in specialized reasoning tasks. LPFQA provides a robust, authentic, and discriminative benchmark for advancing LLM evaluation and guiding future model development.

## 1 Introduction

The rise of Large Language Models (LLMs) has been one of the most significant breakthroughs in the field of artificial intelligence over the past decade, impacting areas such as question answering Zhuang et al. (2023); Li et al. (2024b), reasoning Havrilla et al. (2024); Wang et al. (2023), code optimization Nam et al. (2024); Gu (2023); Fakhoury et al. (2024), and beyond. The ability of LLMs to handle complex tasks has enabled many previously unattainable applications, facilitating their rapid integration into both daily life and professional domains Yang et al. (2024); Zheng et al. (2025). As model architectures and training strategies continue to advance, the accurate and comprehensive evaluation of their true performance becomes increasingly crucial. The current approach involves employing benchmark tests, which are datasets composed of carefully designed questions or tasks. LLMs are required to generate answers or complete these tasks, and their performance is then quantitatively assessed based on the outcomes Chang et al. (2024).

Given that a substantial portion of knowledge in the real world follows a long-tail distribution, which is often fragmented and highly professional, an effective evaluation benchmark should include such long-tail knowledge that is relatively underrepresented in pre-training data Zhang et al. (2023); Yang et al. (2022). Moreover, these questions must be grounded in real-world authenticity to better reflect actual user needs. However, existing benchmarks exhibit clear limitations. For instance, MMLU focuses primarily on simple question answering or multiple-choice tasks, which fail to evaluate a model's ability to handle complex, multi-step reasoning Wang et al. (2024); Hendrycks et al. (2021); HLE Phan et al. (2025) leverages human annotations to approximate human preferences, but its task scenarios are often overly idealized or uncommon, thus not representative of typical user demands. And Arena-Hard Li et al. (2024a), although capturing certain aspects of real user queries, suffers from limited diversity in question types and insufficient difficulty, making it less effective in differentiating performance among LLMs.

To this end, we constructed a comprehensive evaluation benchmark (LPFQA) based on highly professional forums, which characterizes both real-world and long-tail knowledge. The data is collected from technical forums across multiple professional domains. This ensures that tasks of LPFQA are

highly professional, as they are based on complex questions raised by real practitioners with expertise in various fields. At the same time, the data is authentic, as it reflects the real needs and challenges encountered by users in practice. We completed this benchmark construction through three main phases, including (1) data collection and preprocessing, (2) automated question generation and quality control, and (3) expert verification and difficulty adjustment, ensuring that all selected questions fulfill the demands of the benchmark. LPFQA spans 20 academic fields, including Computer Science, Mathematics, Biology, Physics, etc., with a total of 505 questions. We evaluated LPFQA using 12 mainstream models, including GPT, Gemini, DeepSeek, Seed, Qwen, Grok, Claude, and Kimi.

This work introduces LPFQA, an authentic, structured, and interdisciplinary dataset with long-tail knowledge for evaluating LLMs' ability in complex reasoning, providing a robust benchmark for assessing and advancing LLM performance in real-world professional contexts. The main innovations of LPFQA and contributions of this work can be summarized as follows.

- **Innovated evaluation dimension design**. We design a set of fine-grained evaluation dimensions, including knowledge depth, reasoning ability, terminology comprehension, and contextual analysis, ensuring LPFQA 's comprehensiveness in evaluating LLMs' capabilities in handling complex tasks.
- **Hierarchical difficulty design with guaranteed uniqueness**. We employ a tiered difficulty structure to match varying capabilities of different LLMs, while ensuring semantic clarity and answer uniqueness for each task, enhancing the reliability, fairness, and discriminative power of LPFQA.
- **Authentic professional scenario modeling**. We ground questions in authentic use cases by constructing detailed user personas and realistic contextual scenarios, enhancing the ability of LPFQA to validate the performance of LLMs in real-world professional environments.
- **Interdisciplinary knowledge integration**. We integrate long-tail knowledge from diverse fields, improving the LPFQA's effectiveness in evaluating LLMs' integrative capabilities of judgment and reasoning in complex scenarios.

## 2 RELATED WORK

The field of large language model evaluation has seen a rapid proliferation of benchmarks, each designed to probe different facets of model capabilities. Early benchmarks, such as GLUE Wang et al. (2018) and SuperGLUE Wang et al. (2019), focused on a broad range of general language understanding tasks, including question answering and natural language inference. While these benchmarks were instrumental in driving early progress, they are now often considered insufficient for evaluating the nuanced reasoning and vast knowledge base of modern, more capable LLMs. Subsequent benchmarks, such as MMLU Wang et al. (2024), BIG-bench Srivastava et al. (2023), and HELM Liang et al. (2022), extended evaluation to multi-disciplinary knowledge, reasoning, and holistic dimensions of safety, robustness, and fairness. Despite their contributions, these benchmarks still fall short in capturing the challenges of specialized knowledge and complex reasoning, motivating the exploration of new evaluation paradigms.

### 2.1 LONG-TAIL KNOWLEDGE BENCHMARKS

In the real world, data distributions universally exhibit a long-tail characteristic. This implies that a small number of "head" categories account for a significant portion of the data, while the vast majority of "tail" categories are extremely rare. In the context of LLMs, such a distributional imbalance is crucial because the large pre-training corpora, while massive, often lack sufficient coverage of this rare, specialized, or infrequently mentioned "tail" knowledge. As a result, while LLMs demonstrate robust performance on common topics, their ability to handle this long-tail information can decline significantly.

To assess a model's capabilities on long-tail knowledge, researchers have designed specialized benchmarks. The construction methods for these benchmarks primarily fall into two categories: the first is natural data collection, where data is obtained directly from the real world. An example is biodiversity datasets (e.g., iNaturalist Van Horn et al. (2018)), where a large number of species

have very few image samples. This approach captures the most authentic distributions, but data collection is often costly. The second method is synthetic construction, where long-tail distributions are artificially created by imbalanced sampling from existing, balanced datasets (e.g., ImageNet-LT from ImageNet Liu et al. (2019)). While this method is straightforward, it may not fully simulate the complexity and diversity of real-world long-tail data. Although the above benchmarks lay a foundation for evaluating long-tail knowledge, their tasks are often overly simplistic or confined to a few specific domains Liang et al. (2022). These limitations underscore the necessity of developing complementary benchmarks.

## 2.2 USER-CENTRIC AND CHALLENGING BENCHMARKS

In contrast to static long-tail knowledge evaluation, another important class of evaluation methods focuses on a model's performance on dynamic tasks. Chatbot Arena Chiang et al. (2024), for example, is a crowdsourcing platform that evaluates model performance through user blind testing. Its core idea is to have users engage with two anonymous LLMs and vote for the one that performs better. This method effectively captures user preferences and measures a model's overall performance in open-ended conversations. However, crowdsourced evaluation methods like Chatbot Arena also have clear limitations. First, they lack control over specific difficulty or expertise levels. User-submitted questions can be too simple, leading to similar responses from all top-tier models, which makes the benchmark less discriminative. For instance, Arena-Hard Li et al. (2024a) aims to address this issue with adversarial questioning, but its question types can still be relatively concentrated, making it difficult to fully assess a model's capabilities on a broader range of complex, professional long-tail knowledge.

To further test the limits of a model, the Humanity's Last Exam (HLE) Phan et al. (2025) has emerged. HLE is designed to test an LLM's general intelligence and advanced reasoning by collecting extremely difficult questions that even human experts find challenging to answer. These questions typically require cross-disciplinary knowledge integration, complex logical reasoning, and deep comprehension. However, this benchmark also has its limitations. While the questions in HLE are highly challenging, their source and nature may not represent the day-to-day needs of average users. This makes it less effective in evaluating a model's practicality in real-world applications. Furthermore, its extreme difficulty may lead to poor performance from most models, thus limiting its utility as a regular evaluation tool.

Through the analysis above, we recognize the limitations of existing benchmarks. Long-tail knowledge benchmarks lack consideration for complex tasks, while conversational evaluation benchmarks are deficient in terms of domain-specific expertise and difficulty control. Extreme benchmarks like HLE can test a model's cutting-edge capabilities, but their questions have weak relevance to everyday application scenarios. To bridge these gaps, our work aims to construct a new benchmark that can effectively evaluate a model's complex reasoning abilities on professional long-tail knowledge while also reflecting the demands inherent in real-world scenarios.

## 3 LPFQA: LONG-TAIL KNOWLEDGE-BASED BENCHMARK

In this section, we begin with an overview of LPFQA, describing its structure and highlighting its advantages over previous works. Then, we present the detailed steps involved in constructing LPFQA.

### 3.1 OVERVIEW

LPFQA is a long-tail knowledge benchmark, which consists of 505 questions across 20 scientific fields gathered from multiple real professional technical forums, specifically designed for complex reasoning. The following features can distinguish this benchmark.

**Diversity evaluation dimension**. The ability to handle complex tasks is critical for LLMs. To enable the assessment of this ability, LPFQA innovatively covers tasks across multiple evaluation dimensions, including depth of knowledge, reasoning ability, understanding of professional terminology, and contextual analysis.

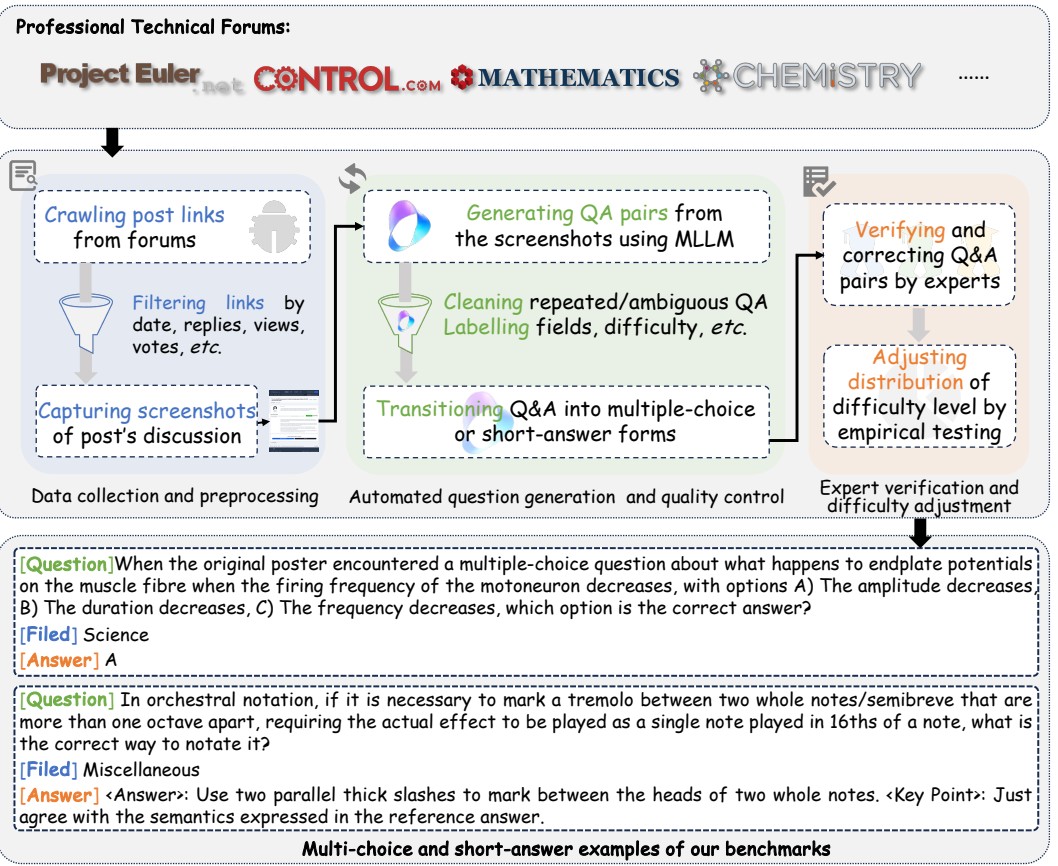

Figure 1: Pipeline of LPFQA's construction

**Discriminative ability and unambiguous guarantee**. To ensure the validity and accuracy of the evaluation results, a benchmark must be discriminative enough to differentiate the abilities of various LLMs, while each task should also be clearly defined. To this end, after careful selection, the tasks in LPFQA can be categorized into distinct levels of difficulty, designed to reflect characteristics suitable for LLMs of varying capabilities. Furthermore, the clarity of each task and the uniqueness of its corresponding answer are guaranteed.

**Derived from real-world scenarios**. To effectively evaluate the response and reasoning capabilities of LLMs in real-world scenarios, a benchmark must closely reflect the types of questions that users genuinely encounter. LPFQA is designed with this objective in mind, emphasizing authentic professional tasks derived from real discussions in technical forums. This design ensures that the tasks are representative of practical situations, thereby enabling a more accurate and realistic evaluation of LLM performance in real-world applications.

**Diversity domains knowledge**. Moreover, LPFQA integrates tasks from a broad spectrum of professional technical forums, spanning domains such as biology, finance, materials science, and computer science. This cross-disciplinary benchmark challenges LLMs to demonstrate comprehensive judgment and reasoning across diverse and complex scenarios.

## 3.2 CONSTRUCTION OF LPFQA

This work develops a fully automated pipeline for constructing such an authentic cross-disciplinary benchmark from professional technical forums. In detail, the whole construction consists of eight steps: ❶ collecting professional forums, ❷ scraping discussion links, ❸ capturing screenshots of discussions, ❹ generating questions from the screenshots using MLLMs, ❺ cleaning up duplicated and ambiguous items with LLMs, ❻ transitioning them into multiple-choice or short-answer form,

❼ verifying all questions by professional experts, and ❽ filtering questions by difficulty through empirical testing, finally.

These steps can be divided into three phases: data collection and preprocessing, automated question generation and quality control, and difficulty adjustment and expert review. This three-phase process follows the natural progression of building a benchmark from raw data to a standardized and high-quality benchmark, ensuring both scalability and reliability.

### 3.2.1 DATA COLLECTION AND PREPROCESSING

**The first phase addresses the challenge of sourcing diverse and representative raw materials**. We manually selected and crowd-sourced several professional forums that represent different disciplines, ensuring coverage across domains such as biology, finance, materials science, and computer science (❶). We developed a customized web crawler to collect forum data at scale. The crawler is capable of adapting to heterogeneous forum structures and supports filtering by metadata such as time, view count, reply count, and vote count, which helps control both the quality and relevance of the collected data (❷). To facilitate later multi-modal content analysis, automated scripts visited each post page and captured screenshots in addition to extracting textual content. This process not only preserved contextual and visual information but also provided a reliable basis for subsequent processing (❸).

### 3.2.2 AUTOMATED QUESTION GENERATION AND QUALITY CONTROL

**The second phase focuses on transforming raw forum content into structured question–answer pairs**. The MLLM first examined each screenshot to determine whether it contained a valid question. Screenshots without valid questions were discarded, while those with valid content proceeded to the next stage. If a post included meaningful replies, the model extracted both the question and key responses to form candidate question–answer pairs; otherwise, only the question itself was retained (❹).

These items then underwent automated quality control with the aid of an LLM. The process included duplicate removal, filtering of incomplete or ambiguous entries, and marking with labels such as domain, clarity, and difficulty. Logical consistency was also checked to ensure alignment between questions and their corresponding answers (❺).

Finally, the validated question–answer pairs were transmitted into multiple-choice or short-answer format. For multiple-choice items, the LLM generated distractor options designed to resemble common errors or misconceptions. For short-answer items, in addition to the correct reference answer, a set of key knowledge points was also provided, which serves as the criterion for determining whether a response is correct. This transition enhanced the usability of the dataset while maintaining both clarity and evaluation effectiveness (❻).

### 3.2.3 EXPERT VERIFICATION AND DIFFICULTY ADJUSTMENT

**The third phase ensures that the question bank achieves a balanced level of difficulty and scientific correctness.** First, the generated items underwent a human verification by the professional experts. They verify the factual accuracy, relevance, and difficulty of each item, while also correcting residual errors introduced during the automated pipeline. This operation enhanced the scientific rigor and reliability of our benchmark (❼).

Finally, to improve the benchmark's ability to differentiate LLMs' capabilities, we conduct an empirical difficulty test. Multiple LLMs were employed to answer all questions, and their accuracy rates were recorded to classify the items into different difficulty levels. The dataset was adjusted by selectively adding or removing items, ensuring a well-balanced difficulty structure (❽).

By integrating the above steps, namely data collection and preprocessing, automated question generation with quality control, and difficulty adjustment with expert review and empirical test-based evaluation, the proposed pipeline achieves end-to-end automation while maintaining high standards of reliability and evaluation utility. This design provides a scalable and systematic approach for constructing a question dataset that faithfully represents real-world professional discourse and is well-suited for LLM evaluation.

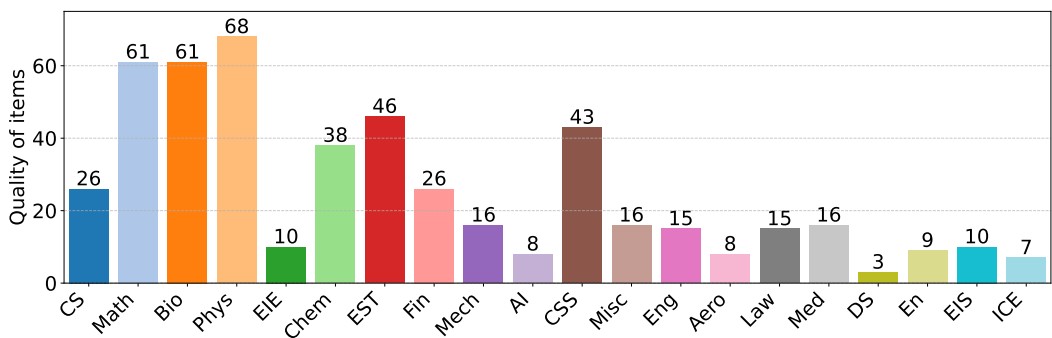

Figure 2: Quality distribution of each field in LPFQA

Table 1: Performances of different models on LPFQA.

| Models | Score |
|---|---|
| Qwen-3 | 38.78 |
| Grok-4 | 39.04 |
| DeepSeek-R1 | 38.25 |
| Seed-1.6 | 41.50 |
| Gemini-2.5-Pro | 44.42 |
| GPT-4.1 | 38.31 |
| GPT-4o | 32.40 |
| o3-high | 43.03 |
| Claude-4 | 38.05 |
| GPT-5 | **47.28** |
| Kimi-K2 | 35.26 |
| DeepSeek-V3 | 32.60 |
| **Average** | 39.08 |

Table 2: Scores of different models on filtered LPFQA.

| Models | LPFQA $^-$ | LPFQA $^=$ |
|---|---|---|
| Qwen-3 | 44.65 | 42.62 |
| Grok-4 | 44.95 | 42.37 |
| DeepSeek-R1 | 44.04 | 41.89 |
| Seed-1.6 | 47.78 | 45.84 |
| Gemini-2.5-Pro | 51.15 | 49.64 |
| GPT-4.1 | 44.11 | 42.45 |
| GPT-4o | 37.31 | 35.03 |
| o3-high | 49.54 | 48.10 |
| Claude-4 | 43.81 | 41.57 |
| GPT-5 | **54.43** | **53.11** |
| Kimi-K2 | 40.60 | 38.58 |
| DeepSeek-V3 | 37.54 | 35.59 |
| **Average** | 44.99 | 43.07 |

### 3.3 STATISTICS OF LPFQA

As depicted in Figure 2, LPFQA covers 20 academic fields with a total of 505 questions, including *Computer Science* (CS), *Mathematics* (Math), *Biology* (Bio), *Physics* (Phys), *Electronic Information Engineering* (EIE), *Chemistry* (Chem), *Electronic Science and Technology* (EST), *Finance* (Fin), *Mechanical and Automation* (Mech), *Artificial Intelligence and Machine Learning* (AI), *Computer Systems and Software* (CSS), *Miscellaneous* (Misc), *General Engineering* (Eng), *Aerospace* (Aero), *Law*, *Medical* (Med), *Data Science and Big Data Technology* (DS), *Energy* (En), *Electronics and Information Science* (EIS), and *Information and Communication Engineering* (ICE). Among them, *Physics*, *Mathematics*, and *Biology* contain the largest number of items, each exceeding 60, while most of the other fields fall within the 10–50 range, and the field of *Data Science and Big Data Technology* has a relatively smaller number, with 3 items.

## 4 EXPERIMENTS

Based on LPFQA, we evaluate the following mainstream models: Qwen-3-235B Yang et al. (2025), Grok-4 xAI (2025), DeepSeek-R1 Guo et al. (2025), Seed-1.6-Thinking Volcengine (2024), Gemini-2.5-Pro Comanici et al. (2025), GPT-4.1 OpenAI (2024a), GPT-4o OpenAI (2024b), o3-high OpenAI (2024c), Claude-4-Sonnet Anthropic (2024), GPT-5 OpenAI (2025), Kimi-K2 Team et al. (2025), and DeepSeek-V3 Liu et al. (2024). All results provided are averaged over three trials.

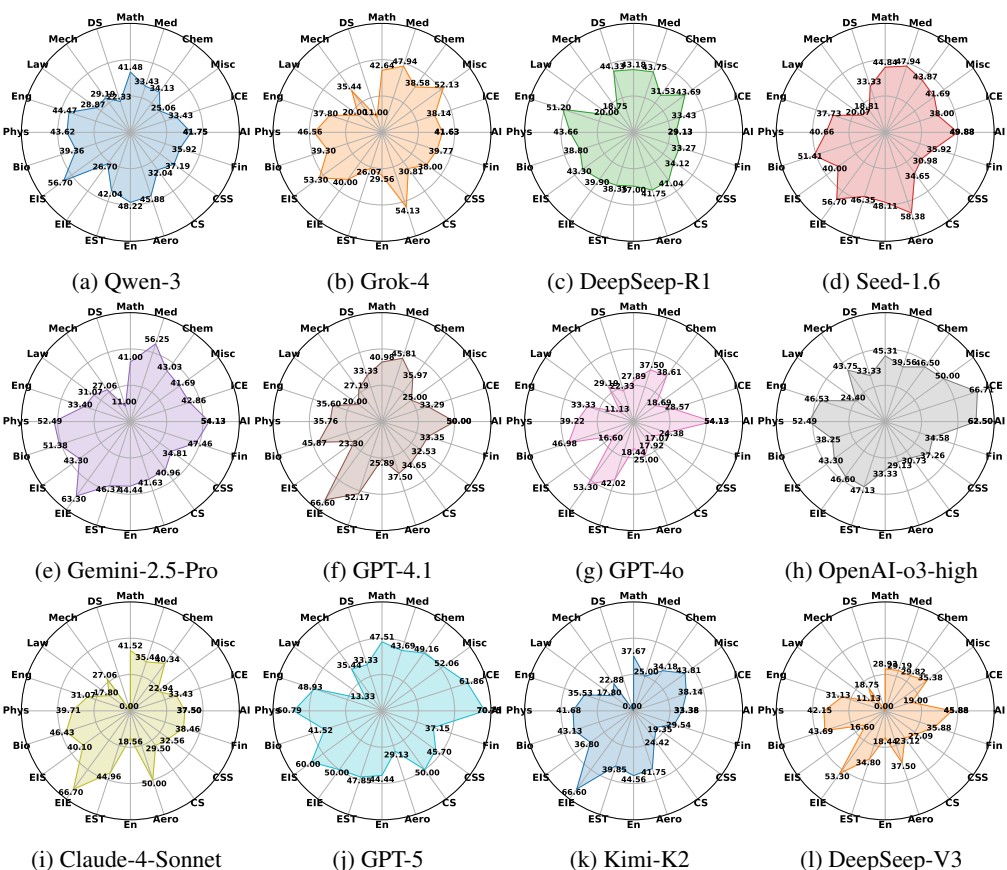

Figure 3: Scores of models on different fields of LPFQA.

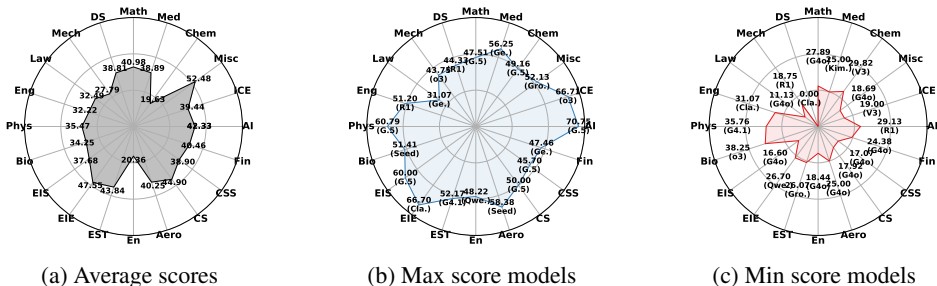

Figure 4: Average, Maximum, and Minimum scores.

## 4.1 MAIN RESULTS

As shown in Table 1, the performance of the evaluated models on LPFQA falls within a relatively narrow range, with scores spanning from 32.40 to 47.28. Among them, GPT-5 achieves the highest score, while GPT-4o records the lowest. To provide a more fine-grained comparison, Figures 3 report the scores of individual models across different fields, offering a clearer picture of their strengths and weaknesses in specific areas. The overall average performance of all models is further summarized in Figure 4a, which provides a holistic perspective on their general capability across fields. Finally, to highlight the comparative extremes, Figures 4b and 4c identify the models that achieve the maximum and minimum scores in each field, thereby providing an intuitive view of their relative advantages and limitations.

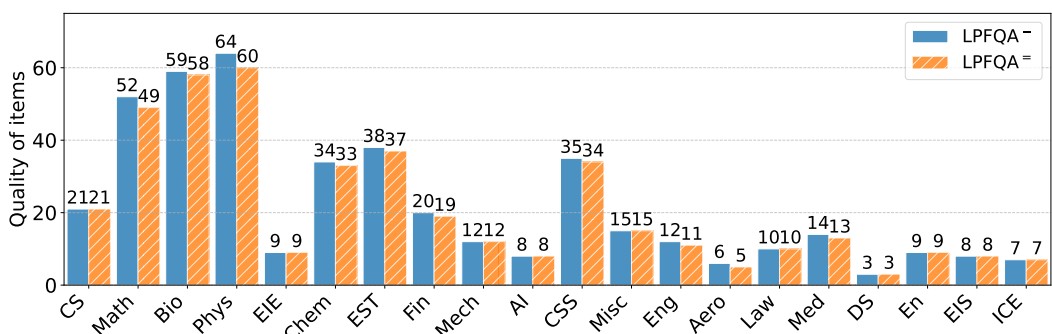

Figure 5: Quality distribution of each field in filtered LPFQA

Based on the results presented in Figures 3 and 4, we analyze the performances of these models from three perspectives: overall performance, disciplinary distribution, and extreme values across models.

- **Overall performance**. Among all evaluated systems, DeepSeek-V3 demonstrates the most balanced and consistent performance across disciplines, with no apparent weaknesses, and can thus be regarded as the overall best-performing model. GPT-5 exhibits strong competitiveness, achieving the highest scores in several domains such as AI, Phys, EIS, Chem, Fin, and CSS, in some cases surpassing DeepSeek-V3. Seed-1.6 and GPT-4.1 also achieve competitive results in specific domains (e.g., CS, Aero, Bio for Seed-1.6; EIT, En for GPT-4.1), though their overall performance remains less comprehensive. Other models, such as Claude-4-Sonnet, Grok-4, and Kimi-K2, tend to show domain-specific strengths but also exhibit noticeable weaknesses, limiting their overall robustness.

- **Disciplinary perspective**. From a disciplinary perspective, clear differences emerge across fields. As shown in Figure 4a, Misc yields the highest average scores (above 50), while En records the lowest overall average (around 20). Other relatively strong domains include Chem, AI, Fin, CS, and EIS, while weaker performance is observed in Med, Law, Eng, and Bio. Intra-model variation is also significant. For example, DeepSeek-R1 attains leading scores in DS, Math, Eng, and Law, but remains comparatively weak in ICE. Similarly, GPT-5 shows clear superiority in Phys and AI, while its performance in Law is less competitive. These disparities indicate that current models continue to face challenges in achieving uniform cross-disciplinary generalization.

- **Max and Min scores**. To provide a comprehensive view beyond average performance, we examine maximum and minimum scores across all disciplines (Figures 4b and 4c). For maximum scores: AI, Phys, EIS, Chem, Fin, and CSS are led by GPT-5; CS, Aero, and Bio by Seed-1.6; DS, Math, Eng, and Law by DeepSeek-R1; EIT and En by GPT-4.1; EIE by Claude-4-Sonnet; ICE by OpenAI-o3-high; and Misc by Grok-4. For Minimum scores: GPT-4o accounts for the lowest performance in multiple domains (Math, Chem, Fin, CSS, CS, Aero, En, and EIS). Other models show more localized weaknesses: Claude-4-Sonnet in DS and Eng, DeepSeek-R1 in Mech and ICE, OpenAI-o3-high in Bio, Qwen-3 in EIT, Grok-4 in EIE, Kimi-K2 in Med, and DeepSeek-V3 in Misc.

## 4.2 DETAIL ANALYSIS

### 4.2.1 FILTERED LPFQA

During our analysis, we observed that none of the evaluated models could correctly answer a subset of questions. Since one of the primary purposes of the benchmark is to differentiate the capabilities of different models, these questions provide little discriminatory value. Therefore, we first excluded them from LPFQA, leaving a remaining set of 436 items. This filtered version, denoted as LPFQA$^-$, was then used to recalculate the distribution of questions across different fields (Figure 5) and the corresponding scores of each model (Table 2).

Table 3: Configured with code interpreter tool

| Models | Score | $\Delta$ |
|--------|-------|----------|
| Qwen-3 | 35.89 | 2.89%↓ |
| DeepSeek-R1 | 34.46 | 3.79%↓ |
| Seed-1.6 | 36.85 | 4.65%↓ |
| Gemini-2.5-Pro | 34.46 | 9.96%↓ |
| GPT-4.1 | 36.12 | 2.19%↓ |
| GPT-4o | 30.28 | 2.12%↓ |
| o3-high | 42.76 | 0.37%↓ |
| GPT-5 | 48.01 | 0.73%↑ |
| Kimi-K2 | 36.12 | 0.86%↑ |
| DeepSeek-V3 | 28.42 | 4.18%↓ |
| **Average** | 36.15 | 7.75%↓ |

Table 4: Configured with search tool

| Models | Score | $\Delta$ |
|--------|-------|----------|
| Qwen-3 | 23.31 | 15.47%↓ |
| DeepSeek-R1 | 33.60 | 4.65%↓ |
| Seed-1.6 | 37.58 | 3.92%↓ |
| Gemini-2.5-Pro | 35.19 | 9.23%↓ |
| GPT-4.1 | 36.32 | 1.99%↓ |
| GPT-4o | 32.60 | 0.20%↑ |
| o3-high | 42.71 | 0.32%↓ |
| GPT-5 | 45.18 | 2.10%↓ |
| Kimi-K2 | 35.52 | 0.26%↑ |
| DeepSeek-V3 | 28.08 | 4.51%↓ |
| **Average** | 35.01 | 10.64%↓ |

In addition, we identified another subset of questions that were answered correctly by all models without exception. While such questions may reflect fundamental or widely shared knowledge, they also contribute minimally to distinguishing the relative strengths and weaknesses of the models. To further emphasize the performance gaps, we excluded these universally solvable questions based on LPFQA$^-$, resulting in a remaining set of 421 items. This second filtered version is denoted as LPFQA$^=$, on which we recomputed both the distributions across different fields (Figure 5) and the model scores (Table 2).

### 4.2.2 ABLATION ANALYSIS

**Does LPFQA evaluate knowledge or reasoning ability?**

We investigated the effect of integrating a Jupyter Code Interpreter (CI) into the reasoning process, which is expected to enhance reasoning ability through code execution. However, as shown in Table 3, it can be observed that overall performance decreased: the scores dropped on most models, and the few improvements that appeared were marginal, leading to a lower overall average. These findings suggest that LPFQA primarily reflects a model's mastery of domain knowledge rather than its reasoning ability.

**Is deep-search always rewarding?**

We incorporated GoogleSearch and TextBrowserView tools into the reasoning process to enable information retrieval. As shown in Table 4, the scores of most models decreased under this setting. We attribute this phenomenon to the nature of LPFQA, which consists of long-tail knowledge that is inherently difficult to retrieve from the web. In such cases, the additional retrieval functions may introduce misleading information during the reasoning process, thereby reducing overall inference accuracy. In other words, for tasks involving long-tail knowledge, simply augmenting models with online search does not provide a positive effect and may even be detrimental. This observation offers valuable insights into the limitations faced by all models when dealing with long-tail knowledge.

## 5 CONCLUSION

In this work, we proposed LPFQA, a long-tail professional forum-based benchmark designed to evaluate LLMs on complex reasoning and specialized knowledge across 20 domains. LPFQA emphasizes authenticity, interdisciplinarity, and fine-grained evaluation dimensions, with hierarchical difficulty and expert verification ensuring reliability and fairness. Our experiments on 12 mainstream LLMs reveal notable disparities, highlighting the persistent challenge of long-tail knowledge. Furthermore, ablation studies show that LPFQA primarily reflects domain knowledge mastery, and that direct integration of external tools does not always enhance performance. Overall, LPFQA provides a robust, discriminative, and authentic benchmark that not only measures current model capabilities but also guides future research toward more generalizable and reliable LLMs.

## ETHICS STATEMENT

This study is based on publicly available professional forum data, which was collected, filtered, and processed in compliance with relevant ethical standards. No personally identifiable or sensitive information was included in the benchmark. All data used were anonymized and only retained for research purposes. The benchmark construction and experiments were conducted strictly for academic evaluation and model analysis, without any intention of infringing on privacy, spreading harmful content, or causing potential misuse. We affirm that this research adheres to the ethical principles of fairness, transparency, and responsible AI development.

## REPRODUCIBILITY STATEMENT

To foster transparency and facilitate reproducibility, we will release our benchmark to the public. Furthermore, we provide the details of the benchmark construction process in the appendix, including: (1) all prompts used for question generation, (2) the prompts applied for evaluation criteria, and (3) the complete list of forums utilized. We believe these resources will enable the community to faithfully reproduce our results and build upon our work.

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

# APPENDIX

## A LLM USAGE STATEMENT

In the preparation of this manuscript, we employed LLMs solely for textual polishing and language refinement. The motivation, research design, etc., were independently conducted by the authors.

## B EXAMPLES Q&A OF LPFQA

---

**Q&A 1, Field:** *General Engineering*

**Question**: When only 110V service is available for a Millport milling machine with a 220V single-phase motor, which power supply solution is recommended, and what key factor must be considered for selecting this equipment?
A. A voltage regulator with adjustable output, focusing on maximum current capacity alone.
B. A three-phase to single-phase converter with 110V input, needing to match the motor speed rating.
C. A Variable Frequency Drive (VFD) with single-phase input and three-phase output, requiring matching the motor's horsepower (HP) rating and current requirements.
D. A step-up transformer with single-phase input and single-phase output, requiring matching the voltage ratio only.
E. A capacitor-start motor conversion kit, requiring compatibility with motor phase configuration.
F. A DC power supply with inverter function, needing to match the motor frequency range.
**Answer**: C

---

---

Q&A 2, Field: *Miscellaneous*

**Question**: According to Einstein's theories and current research, which of the following correctly identifies a fundamental challenge to light-speed space travel and a promising direction to overcome such obstacles?
A. Gravitational field distortion near light speed; testing warp drive prototypes
B. Radiation shielding limitations at relativistic speeds; deploying solar sail technology
C. Infinite energy requirement due to relativistic mass increase; developing nuclear propulsion systems
D. Time dilation effects causing crew aging issues; advancing ion thruster technology
E. Quantum vacuum fluctuations disrupting navigation; upgrading chemical rocket efficiency
F. Infinite energy requirement due to relativistic mass increase; improving chemical propulsion methods
**Answer**: A, B, C, F

---

Q&A 3, Field: *Biology*

**Question**: According to the Wright Fisher model of population genetics, please calculate how many generations of living offspring would an average person typically have before the extinction of their genetic lineage?
A. Approximately 10 generations
B. Approximately 20 generations
C. Approximately 25 generations
D. Approximately 30 generations
E. Approximately 40 generations
F. Approximately 60 generations
**Answer**: B

---

Q&A 4, Field: *Physics*

**Question**:Which of the following accurately describes stellar rotation and the key physical factors determining rotation rate and axis orientation?
A. Not all stars rotate, with key determinants including stellar mass, angular momentum loss during evolution, binary interactions, planetary system formation, and magnetic activity effects
B. All stars rotate, with key determinants including angular momentum loss through stellar winds, binary interactions, planetary system formation, and magnetic activity effects
C. Not all stars rotate, with key determinants including initial angular momentum from the parent molecular cloud, angular momentum loss during evolution, binary interactions, planetary system formation, and stellar radius fluctuations
D. Not all stars rotate, with key determinants including initial angular momentum from the parent molecular cloud, angular momentum loss during evolution, binary interactions, planetary system formation, and magnetic activity effects
E. All stars rotate, with key determinants including initial angular momentum from the parent molecular cloud, nuclear fusion efficiency, and binary interactions
F. Not all stars rotate, with key determinants including initial angular momentum from the parent molecular cloud, angular momentum loss during evolution, binary interactions, and magnetic activity effects
**Answer**: D

## C    PROMPTS OF LLM AS A JUDGE

This benchmark was generated with the use of MLLM and LLM, and the relevant steps involved the following prompts.

702
703

**Prompt for image to Q&A**

704
705
706
707

You are a professional expert in data scraping and question generation. Your primary responsibility is to identify the content from forum post screenshots, accurately extract the original poster's question and others' replies, and then generate high-quality questions with a certain level of difficulty and misleading elements based on the scraped content.

708 Task Steps:

709 Step 1: Page Validity Check

710 1. Determine whether the current screenshot is a valid discussion page (not a 404 page,
711 blank page, login page, or advertisement page) 2. If it is an invalid page, return directly:
712 "is_valid": false

713 Step 2: Question Validity Check

714 3. If the page is valid, check whether it contains a clear question
715 4. If no valid question is found (e.g., title only without content, meaningless characters, or
716 simple informational posts without questions), return: "is_valid": false

717 Step 3: Content Extraction and Processing

718 5. If there is a valid question and valid replies (substantial content, not just emojis or meaningless characters)

719 6. Extract the title, question description, and all relevant reply content to form a structured
720 triplet: (question, answer, context)

721 Step 4: Result Generation

722 7. Based on the extracted core content, create high-quality academic questions that meet the
723 following criteria:

724 - The question must be self-contained and independent of external context

725 - Rephrase the question using clear and natural language to enhance clarity, fluency, and
726 understandability, avoiding ambiguity or vague expressions

727 - Be sufficiently professional — not simple knowledge quizzes or common-sense questions,
728 but those requiring logical reasoning and analysis

729 - Require multi-step, cross-disciplinary complex reasoning

730 - Involve specialized or cutting-edge knowledge in specific domains, not purely general
731 academic knowledge

732 - Avoid simple factual questions; emphasize comprehension and application

733 - The answer must be unique and verifiable, capable of being confirmed through reasoning or
734 experimentation; exclude open-ended questions with multiple possible answers or solution
735 paths

736 - Use the same language as the original forum post for the question and answer (e.g., if the
737 post is in English, both question and answer should be in English) 8. Summarize and organize the most accurate, comprehensive, and detailed answer from the replies (leave blank if
738 no valid reply exists)

739 9. Provide a detailed explanation and reasoning process for the answer (leave blank if no
740 valid reply exists)

741 10. Set the "has_answer" flag: 1 if there are valid replies and an answer can be extracted, 0
742 otherwise

743 Please strictly return the result in the following JSON format without adding any extra text:

```
{
    "is_valid": boolean,
    "question": string,
    "answer": string,
    "explanation": string,
    'has_answer': integer
}
```

752
753

**Prompt for quality filter**

754
755

Task: Evaluate the quality of the following QA pair and retain only high-quality questions
Evaluation Criteria (all must be satisfied):

- Accuracy: All background knowledge, data, and facts involved in the question are accurate and error-free
- Completeness: The question provides all information necessary for solving it, with no missing or redundant information
- Professionalism: The question is sufficiently professional—not a simple knowledge quiz or common-sense question—but one that requires logical reasoning and analysis to solve
- Timelessness: The question does not involve time-sensitive content; it remains valid and true at all times
Difficulty Criteria: The question must meet the following requirements:
- Depth of Knowledge: Solving the question requires specialized or cutting-edge knowledge in a particular field
- Length of Reasoning Chain: The solution process involves multi-step, cross-disciplinary complex reasoning
- Abstraction and Synthesis Requirements: The question demands high-level abstract thinking and information synthesis
- Deceptiveness and Originality: The question requires creative or unconventional problem-solving approaches
QA pair to be evaluated:
    Question: {question}
    Answer: {answer}
    Explanation: {explanation}
Please return only the result in JSON format, including the following fields:
- is_qualified: boolean (whether the question meets the standards)
- reason: string (reason for not meeting the standards; empty string if qualified)

---

### Prompt for multi-choice

Task: Convert the question, answer, and explanation from the following QA pair into a high-quality multiple-choice question
Requirements:
1. Retain the core of the original question, but adjust the phrasing appropriately to suit the multiple-choice format
2. Generate 5 to 7 options, with exactly one correct answer and the rest as plausible distractors
3. Distractors should be misleading and belong to the same knowledge domain as the correct answer
4. Options must be randomly ordered
5. Ensure all options have a consistent grammatical structure
6. Maintain a high difficulty level, requiring multi-step reasoning to solve
7. The language used must be consistent with the original question—if the original QA pair is in English, the output must also be in English; do not change the language arbitrarily
QA pair:
    Question: {question}
    Answer: {answer}
    Explanation: {explanation}
Please return only the result in JSON format, including the following fields:
- question: string (the multiple-choice question)
- options: array[string] (list of options, including one correct answer)
- correct_answer: string (the content of the correct answer)
- explanation: string (explanation of why this option is correct)

---

### Prompt for short answer

Task: Convert the question, answer, and explanation from the following QA pair into a high-quality short-answer question, and extract the core knowledge points
Requirements:

1. Retain the core of the original question, with appropriate rephrasing to better suit the short-answer format
2. Generate a detailed and comprehensive answer that includes all key information from the original explanation
3. The answer must be substantial, well-supported, and logically structured, avoiding vague or generic statements
4. Use natural, fluent, professional, and accurate language
5. Maintain a high difficulty level, reflecting professional knowledge and reasoning processes
6. Summarize 1–5 core knowledge points—these are the key concepts essential for determining the correctness of the answer
7. The language used must be consistent with the original question—if the original QA pair is in English, the output must also be in English; do not change the language arbitrarily
QA pair:
    Question: {question}
    Answer: {answer}
    Explanation: {explanation}
Please return only the result in JSON format, including the following fields:
- question: string (the short-answer question)
- answer: string (detailed answer, incorporating the explanation)
- core_knowledge_points: array[string] (list of core knowledge points)

---

**Prompt of metric for answer evaluation**

From now on, your role is that of a professional grading teacher. Your task is to objectively score the ¡Student Answer¿ based on the ¡Reference Answer¿ and ¡Evaluation Points¿ that I provide. The main steps and rules include the following 6 points:

1. Scoring Levels: There are only two levels, 0 and 1. No other scores are allowed, so please do not give any score other than 0 or 1.

2. Since I have not provided you with the ¡Question¿, you only need to judge whether the content of the ¡Student Answer¿ meets the ¡Evaluation Points¿. Do not imagine or infer the content of the ¡Question¿. Note that ¡Evaluation Points¿ take precedence over the ¡Reference Answer¿. If there is a conflict between them, you must prioritize the requirements of the ¡Evaluation Points¿.

3. You need to first determine whether the ¡Student Answer¿ meets the requirements of the ¡Evaluation Points¿ (if no ¡Evaluation Points¿ are provided, then by default, the results and requirements in the ¡Reference Answer¿ are considered as the evaluation points). The answer must meet all conditions in the evaluation points. If any one of them is not satisfied, you should immediately give a score of 0, stop executing Step 4, and output the result according to the format in Step 5. If all conditions are met, continue to Step 4.

4. Determine whether the final answer in the ¡Student Answer¿ is correct:
   - You do not need to consider whether the process is correct, only the final answer.
   - If the final answer is identical or synonymous to the ¡Reference Answer¿ (e.g., "two times" and "2"), then give a score of 1. Otherwise, if it is inconsistent or not synonymous, give a score of 0.
   - Some answers may contain errors, but if later self-corrected, the final result after correction should be regarded as the actual answer.
   - Some answers may include self-analysis or feedback, possibly repeating outputs multiple times. Do not be misled by intermediate outputs—only use the last final result for grading.

5. Format Requirements: Before outputting your result, first check your analysis process and score to ensure they are reasonable and correct. If there are any errors

or omissions, revise them in time. Finally, when you believe everything is correct, provide the output strictly in the following format:

- Grading Basis: (Concise explanation of the grading reason, less than 100 tokens)
- Score: x (output must be "0" or "1")
- JSON:

```
{"answer_score": your score}
```

6. Before you formally start the grading task, to improve your grading accuracy and understanding of the scoring standards, I will first provide a simulated scoring example. By studying this example, you will become more familiar with the grading process and master the grading techniques:

EXAMPLE

¡Reference Answer¿:

5. Move to the kitchen.
3. Find the kettle, check that it has water.
2. Pick up the kettle and move to the balcony.
4. Water the flowers.
Path: 5, 3, 2, 4

¡Evaluation Points¿:

1. The "watering process" in the student's answer must be exactly consistent with the reference answer; otherwise, score 0.

2. The "path" in the student answer must be exactly consistent with the reference answer, otherwise score 0.

¡Student Answer¿:

5. Move to the kitchen.
1. Find a sponge, check if it has water.
6. Pick up the sponge and move to the balcony.
4. Water the flowers.
Path: 5, 1, 6, 4

Grading Basis: Both process and path are inconsistent with the reference answer, hence score 0.
Score: 0
JSON:

```
{"answer_score": 0}
```

I hope you can fulfill the role of grading teacher. If you perform well, I will give you appropriate rewards. Please strictly follow the output format I provide; otherwise, I will penalize you. Also, always use the final result of the answer for grading, and do not be misled by intermediate outputs.

## D  FORUM LIST

The forums selected include, but are not limited to, the following:
https://scienceforums.net/forum/80-sciences/
https://stats.stackexchange.com/
https://math.stackexchange.com/
https://mathoverflow.net/
https://mathematica.stackexchange.com
https://or.stackexchange.com
https://geant4-forum.web.cern.ch/

918 https://root-forum.cern.ch/
919 https://quantumcomputing.stackexchange.com
920 https://www.physicsforums.com/
921 https://astronomy.stackexchange.com
922 https://physics.stackexchange.com
923 https://worldbuilding.stackexchange.com
924 https://chemistry.stackexchange.com/
925 https://crafts.stackexchange.com
926 https://biology.stackexchange.com
927 https://medicalsciences.stackexchange.com/
928 https://bioinformatics.stackexchange.com
929 https://bioacoustics.stackexchange.com
930 https://www.biostars.org/
931 https://space.stackexchange.com
932 https://drones.stackexchange.com
933 https://aviation.stackexchange.com
934 https://eaaforums.org/
935 https://www.eng-tips.com/
936 https://mechanics.stackexchange.com
937 https://engineering.stackexchange.com
938 https://bicycles.stackexchange.com
939 https://3dprinting.stackexchange.com
940 http://www.mjtd.com/
941 https://www.practicalmachinist.com/
942 https://www.practicalmachinist.com/
943 https://diysolarforum.com/
944 https://cr4.globalspec.com/thread/88025/High-Voltage-Engineering
945 https://www.elitetrader.com/
946 https://quant.stackexchange.com/
947 https://patents.stackexchange.com
948 https://law.stackexchange.com/
949 https://answers.justia.com/
950 https://iot.stackexchange.com
951 https://ham.stackexchange.com
952 https://electronics.stackexchange.com
953 https://dsp.stackexchange.com
954 https://arduino.stackexchange.com
955 https://patents.stackexchange.com/
956 https://www.lawanswers.com.au/forums/defamation-law-forum.25/
957 https://3dprinting.stackexchange.com
958 https://android.stackexchange.com
959 https://artofproblemsolving.com/community
960 https://arduino.stackexchange.com
961 https://ai.stackexchange.com
962 https://apple.stackexchange.com
963 https://patents.stackexchange.com
964 https://board.asm32.info/
965 https://aviation.stackexchange.com
966 https://learn.microsoft.com/en-us/answers/topics/azure-digital-twins.html
967 https://alcohol.stackexchange.com
968 https://bioacoustics.stackexchange.com
969 https://bioinformatics.stackexchange.com
970 https://biology.stackexchange.com
971 https://www.biostars.org/
https://bitcoin.stackexchange.com
https://blender.stackexchange.com
http://forums.corvetteforum.com/index.php
https://cardano.stackexchange.com
https://chinese.stackexchange.com

972 https://civicrm.stackexchange.com
973 https://codegolf.stackexchange.com
974 https://computergraphics.stackexchange.com
975 https://cs.stackexchange.com/
976 http://www.cplusplus.com/forum/
977 https://crypto.stackexchange.com
978 https://datascience.stackexchange.com
979 https://dba.stackexchange.com
980 https://discuss.dvc.org/
981 https://electronics.stackexchange.com
982 https://emacs.stackexchange.com
983 https://engineering.stackexchange.com
984 https://ethereum.stackexchange.com
985 https://forum.filezilla-project.org/index.php
986 http://www.fluka.org/fluka.php?id=mailinglist&mm2=6
987 https://french.stackexchange.com
988 https://gamedev.stackexchange.com
989 https://engx.theiet.org/
990 https://iot.stackexchange.com
991 https://forums.majorgeeks.com/
992 https://mattermodeling.stackexchange.com
993 https://community.myfitnesspal.com/en/categories/forums
994 https://networkengineering.stackexchange.com
995 https://opensource.stackexchange.com
996 http://www.openedv.com/
997 https://or.stackexchange.com
998 https://parenting.stackexchange.com
999 https://money.stackexchange.com
1000 https://www.physicsforums.com/
1001 https://pm.stackexchange.com
1002 https://proofassistants.stackexchange.com
1003 https://psychology.stackexchange.com/
1004 https://puzzling.stackexchange.com
1005 https://discuss.pytorch.org/
1006 https://quant.stackexchange.com
1007 https://quantumcomputing.stackexchange.com
1008 https://quantumcomputing.stackexchange.com/
1009 https://forums.raspberrypi.com/
1010 https://www.reddit.com/r/math/
1011 https://root-forum.cern.ch/
1012 https://softwareengineering.stackexchange.com
1013 https://community.spiceworks.com/
1014 https://medicalsciences.stackexchange.com/
1015 https://stackoverflow.com/questions/tagged/robotics
1016 https://www.statalist.org/forums/forum/general-stata-discussion/general
1017 https://stellar.stackexchange.com
1018 https://www.techpowerup.com/forums/
1019 https://tex.stackexchange.com
1020 https://tezos.stackexchange.com
1021 https://unix.stackexchange.com
1022 https://ux.stackexchange.com
1023 https://www.vnpy.com/forum/
1024 http://forums.vwvortex.com/
1025 https://guba.eastmoney.com/
https://bbs.pinggu.org/
http://www.mjtd.com/
http://www.3dportal.cn/
http://www.proewildfire.cn/
https://www.armbbs.cn/

