# OpenReview forum: "LPFQA: A Long-Tail Professional Forum-based Benchmark for LLMs' Evaluation"
_ICLR.cc/2026/Conference — ICLR 2026 Conference Withdrawn Submission_

### Official Review · Reviewer_fdUJ · 2025-10-17

**Soundness:** 1
**Presentation:** 2
**Contribution:** 1
**Rating:** 2
**Confidence:** 4

**Summary:**

This paper introduces LPFQA, a new benchmark designed to evaluate the ability of Large Language Models (LLMs) to handle professional, long-tail knowledge. The benchmark consists of 505 questions gathered from 20 professional forums across different disciplines. Based on tests conducted on 12 mainstream LLMs, the study reveals significant performance disparities among them.

**Strengths:**

1.  The paper accurately identifies a critical gap in current LLM evaluation—the lack of assessment for real-world long-tail knowledge—and innovatively uses professional technical forums as a data source to address this challenge. This approach ensures that the benchmark's questions are grounded in the authentic needs of practitioners, lending them high authenticity.

2.  The  construction process outlined in the paper, which combines automated generation (using MLLMs/LLMs) with expert verification, is systematic and well-defined. This framework provides a valuable and replicable methodology for creating similar benchmarks for other specialized long-tail knowledge domains.

3.   The paper offers valuable preliminary insights through its ablation studies.

**Weaknesses:**

1. The Core Concept of "Long-Tail" is Ill-Defined and Unsubstantiated: The paper's central claim rests on the term "long-tail knowledge," yet this concept is never operationally defined or empirically verified. The authors operate on the unsubstantiated assumption that any question originating from a specialized professional forum automatically qualifies as "long-tail." This is a critical conceptual flaw that lacks evidentiary support.

2. The Data Curation and Verification Process Lacks Critical Transparency: The methodology section (3.2) describes a procedural pipeline but omits the crucial details required to assess its validity and potential biases. Key decisions regarding the selection of forums, the qualifications and instructions for the "professional experts," and measures for inter-rater reliability are completely absent, making the process a "black box" and impossible to scrutinize.

3. The Dataset is Insufficiently Small and the Filtering Logic is Contradictory: With only 505 questions spread across 20 domains, the dataset is too small to support robust conclusions about the capabilities of large-scale models. Furthermore, the decision to filter out questions that all models answered correctly is logically inconsistent with the "long-tail" premise; if a question is truly long-tail, it should not be easily solved by all general-purpose models. This suggests the filtering may be artificially creating difficulty rather than reflecting the data's natural properties.

4. Ablation Studies are Oversimplified and Lead to Unjustified Conclusions: The paper draws strong, overly broad conclusions from simplistic ablation studies. Attributing the failure of a Code Interpreter solely to "reasoning being irrelevant," or the failure of search tools solely to the "nature of long-tail knowledge," ignores numerous significant confounding variables (e.g., model knowledge cut-off dates, poor query generation, the specific type of reasoning being tested).

5. The Evaluation Protocol Has Unaddressed Limitations: The paper fails to discuss the well-documented limitations of its chosen evaluation method, particularly the use of an "LLM-as-Judge" for open-ended questions. It also omits essential information about the final composition of the benchmark (e.g., the ratio of multiple-choice to short-answer questions), hindering a complete interpretation of the results.

**Questions:**

1. On Defining "Long-Tail":
How do you operationally define "long-tail knowledge" in a way that is measurable and falsifiable, beyond simply its origin in a professional forum?
What empirical evidence can you provide to demonstrate that the concepts and questions within LPFQA are indeed "low-frequency" or "rare" within large-scale pre-training corpora?
How does your methodology distinguish between a question that is genuinely "long-tail" (rare knowledge) versus one that is simply "difficult" (requiring complex reasoning about potentially common knowledge)?

2. On Data Curation Transparency:
What were the specific, objective criteria used to select the source forums? This is essential for understanding the dataset's scope and potential biases.
Could you please detail the qualifications of the "professional experts" for each domain? What specific instructions (manual) were they provided for validation, and what was the measured inter-rater reliability?
How do you account for the severe data imbalance across domains (e.g., 68 questions in Physics vs. 3 in Data Science)? Is this an artifact of the collection method or a reflection of the domains themselves?

3. On Dataset Integrity and Filtering:
Given the small and imbalanced sample size, how can you justify making strong claims about a model's specific domain strengths or weaknesses?
If a significant portion of the initial data could be answered correctly by all models, doesn't this challenge the core premise that the data is inherently "long-tail"? What percentage of questions were removed during the "all correct" and "all incorrect" filtering stages?

4. On the Interpretation of Ablation Studies:
Regarding the Code Interpreter experiment, how did you rule out alternative explanations for the performance drop, such as the model's inability to translate non-computational problems into effective code?
For the search tool experiment, how did you control for confounding variables like the models' knowledge cut-off dates or the quality of the LLM-generated search queries?

5. On the Evaluation Protocol:
What steps were taken to validate the reliability of the "LLM-as-Judge," especially for nuanced short-answer questions, and to mitigate its known biases (e.g., preference for verbosity)?
What is the final distribution of multiple-choice versus short-answer questions in the LPFQA benchmark? How might this composition affect the interpretation of the overall model scores?

---

### Official Review · Reviewer_wqkd · 2025-10-29

**Soundness:** 2
**Presentation:** 3
**Contribution:** 2
**Rating:** 2
**Confidence:** 4

**Summary:**

This paper presents LPFQA, a dataset containing 505 questions drawn from professional forums across various fields. The authors motivate their work by highlighting limitations in existing benchmarks and mention that current long-tail benchmarks lack complexity, while conversational benchmarks provide limited control over expertise and difficulty levels. To construct LPFQA, the authors propose a three-stage pipeline for dataset creation involving data collection, question generation, and expert verification. They then evaluate the dataset on 12 large language models and analyze their performance across different domains.

**Strengths:**

- The paper is well-written and easy to follow.
- The dataset is collected from professional forums, offering real-world scientific questions that add practical relevance.
- The number of models used in evaluation is through.

**Weaknesses:**

- The dataset size is limited. Although the authors claim to cover multiple fields, Figure 2 shows that many domains contain fewer than 20 questions, which is insufficient to support the performance evaluations made in Section 4.1.
- While the authors state that the dataset creation process is end-to-end automated and scalable, the expert verification stage introduces a manual expert verification, which limits the overall scalability of the approach.
- The evaluation design has several shortcomings:
  - The paper claims that LPFQA assesses long-tail knowledge, but this concept is not formally defined, nor is it clearly demonstrated through experiments. It would help if the authors clarified whether the ablations in Section 4.2.2 aim to support this claim, and if so, a comparison with existing long-tail benchmarks would strengthen the argument.
  - The paper states that the difficulty level of the dataset is adjusted, but no clear definition or validation of difficulty is provided. It is unclear how this factor influences LLM performance.
  - Multiple-choice and short-answer questions are not analyzed separately, which could obscure insights about model performance across question types.
  - The paper lacks a detailed error analysis, leaving unclear why models fail on questions in this dataset or what types of errors are most common for LPFQA.

**Questions:**

- How do you ensure that the LLM-generated distractors are of high quality and guaranteed to be incorrect (i.e., that each multiple-choice question has only one valid answer)?
- What portion of the dataset consists of multiple-choice questions versus short-answer questions?
- How do you ensure that the LLM does not hallucinate or introduce errors when extracting questions from screenshots?
- Could you provide more details about the experimental setting described in Section 4? Specifically, what temperature was used when prompting the LLMs to answer the questions, and did you experiment with chain-of-thought or few-shot prompting?

---

### Official Review · Reviewer_YZDZ · 2025-10-31

**Soundness:** 1
**Presentation:** 1
**Contribution:** 1
**Rating:** 0
**Confidence:** 4

**Summary:**

This paper proposes a new benchmark dataset that targets at long-tail professional benchmark to evaluate LLMs. It proposes questions from 20 fields and 502 tasks in total. Experiments indicate that even the strongest LLMs still cannot fully solve this task.

**Strengths:**

Long-tail QA benchmark for professional domain, as long as we have clear definition, is an interesting and useful task.

**Weaknesses:**

Unfortunately this is below the bar of this conference.
1. The paper focuses on long-tail QA, but it lacks a clear definition of what constitutes “long-tail.” This concept should be distinguished from cases where an LLM lacks parametric knowledge. In my view, the proposed task does not align well with the common understanding of “long-tail” .
2. The experiments require significant improvement. The current evaluation is only limited to running different LLMs or API-based systems, without providing meaningful insights into why these models fail.
3. Significant amount of this paper is written by LLMs.

**Questions:**

N/A

---

### Official Review · Reviewer_oqiX · 2025-11-03

**Soundness:** 1
**Presentation:** 1
**Contribution:** 1
**Rating:** 0
**Confidence:** 5

**Summary:**

This paper proposes LPFQA, a long-tail benchmark derived from professional forums. A semi-automated pipeline scrapes online forums, captures screenshots of the pages, then tasks an MLLM with extracting and re-writing a QA example, with examples then verified by a human. The authors then baseline ten models on this benchmark, doing simple prompting, and also exploring code-interpreter and search tool-enabled variants.

**Strengths:**

The authors pose an interesting framing of finding a sweet-spot in task difficulty that reflects real-world 'professional' settings, contrasting this with 'head' examples that were likely seen during pre-training and extreme examples that are too challenging (HLE). With that said, I think this significance of this niche target area could be motivated further with fuller comparison to existing works.

**Weaknesses:**

This seems to be a *very* initial draft of this work, and requires significant extra work in both motivation, formulation, and evaluation. Further, the paper requires extensive writing and presentation improvements. The novelty of the contribution is especially questionable -- the authors largely scrape online data without truly justifying if it matches the long-tail distribution they target. In summary, the work needs comprehensive bolstering.

**Questions:**

Some comments and questions that I hope are helpful in improving this work:

- Some domains are extremely small (3 items).
- Why screenshots instead of text during the scraping process? Is this simply to ease the parsing process? If screenshots are required for multimodal context, can quantify the proportion of items that truly require vision?
- Can you better motivate why this particular data reflects the long-tail use case you target? It would be ideal to provide quantitative analysis of this, rather than simple qualitative comparisons with existing works.
- Can you provide full details of your expert annotation process? Did you hire experts from each of the domains? Do you check annotator agreement in any way?
- Is any diversity injected during rewriting, or is the QA re-writing step largely to clean-up the example and map it into text?
- Have you ensure it is legal to scrape all of these websites and use their content for AI-related purposes? I would suggest detailing this clearly in the paper, as many forums now have restrictions.

Presentation / Typos / Grammar Issues:
- **Citation format is incorrect**. Citations should be placed within parentheses.
- Inconsistent totals in dataset size (502 vs. 505).
- Define LFPQA within in the paper text
- Some of your phrasing is incorrect. E.g., “Diversity domains knowledge” → “Diverse domain knowledge”
- Normalize quotation marks and spacing: fix curly vs. straight quotes and stray spaces (e.g., “LPFQA ’s”).
- Simply the radar chart presentation. It is hard to grasp any key takeaways as there is too much information presented.
- Section 3.3 can be moved to the Appendix

I hope this feedback is helpful!

---

### Note · Authors · 2025-11-23

I have read and agree with the venue's withdrawal policy on behalf of myself and my co-authors.